# Enhancing Cross-domain Link Prediction
# via Evolution Process Modeling

## ABSTRACT

This paper proposes CrossLink, a novel framework for cross-domain link prediction. CrossLink learns the evolution pattern of a specific downstream graph and subsequently makes pattern-specific link predictions. It employs a technique called *conditioned link generation*, which integrates both evolution and structure modeling to perform evolution-specific link prediction. This conditioned link generation is carried out by a transformer-decoder architecture, enabling efficient parallel training and inference. CrossLink is trained on extensive dynamic graphs across diverse domains, encompassing 6 million dynamic edges. Extensive experiments on eight untrained graphs demonstrate that CrossLink achieves state-of-the-art performance in cross-domain link prediction. Compared to advanced baselines under the same settings, CrossLink shows an average improvement of **11.40%** in *Average Precision* across eight graphs. Impressively, it surpasses the fully supervised performance of 8 advanced baselines on 6 untrained graphs.

## KEYWORDS

Dynamic Graph, Link prerdiction

## 1 INTRODUCTION

Dynamic graphs are widespread in the real world [6, 43], their nodes representing entities and dynamic edges denoting complex interactions between them [17]. For example, in recommendation systems, users have dynamic interactions with items, forming user-item dynamic graphs [17]. In social networks, interactions between users create user-user dynamic graphs [15]. Consequently, dynamic graph modeling has recently attracted significant attention from the academic community and has become an important branch of graph research [43].

Link prediction (LP) is a crucial task in dynamic graph modeling. In real world, many interaction-related applications can be formed by this task [6, 17]. For example, predicting a user's purchase interest can be represented as an LP on a user-item network [41], where an edge denotes a purchase relationship. Similarly, friend recommendations can be represented as LP on user-user networks [38]. However, current methods mainly consider single graph setting [12]. In this setting, the graph model is trained using supervised learning on a given graph and then makes inferences on the same graph (referred to as *End2End setting*). This approach has several notable limitations when applied in real-world scenarios: (1) High human/time costs: The *End2End setting* requires independently training different models for each graph. Each training process demands careful design and optimization of hyperparameters by experts. Additionally, the training process is time-consuming. (2) Unsuitability for small datasets: The *End-to-End* setting typically requires a substantial number of samples for satisfactory domain-specific performance. This makes it ill-suited for small-scale application scenarios, such as B2B businesses or situations involving large

graphs with limited data. (3) Inability to learn more knowledge from different applications: Graphs in different applications may contain complementary knowledge. For instance, users purchasing items and users listening to music are both projections of human behavior. Therefore, learning from both user-item graphs and user-music graphs can help the model better understand behavior-related knowledge. However, End2End training is limited to a single graph.

Therefore, this paper aims to explore a novel problem: how to perform cross-domain link prediction on dynamic graphs? In this setting, a single graph model is trained once using multiple graphs and then directly applied to predict links on unseen downstream graphs (in this paper, "domain" is equivalent to "graph"). Compared to *End2End setting*, cross-domain link prediction offers several clear advantages: (1) Only one model must be trained, which can be applied across various scenarios/graphs. (2) Since the inference stage does not require additional training, this setting is naturally suitable for scenarios with insufficient training samples. (3) By training on multiple graphs, the model can acquire a broader range of knowledge.

However, cross-domain link prediction faces a fundamental challenge: how to model ambiguous structures. Different graphs are interdependent [12, 42], meaning the same structure may hold different meanings and evolve differently across various graphs. As shown in Figure 1, Graph A typically follows a *triadic closure process* [11, 15, 22, 44], where two nodes with common neighbors are more likely to form edges, while Graph B exhibits a contrasting pattern. Consequently, even if the node pair (red and blue nodes) in Graph A and Graph B has the same local structure, their ground truths are different. We refer to this type of local structure, which has diverse ground truths across various graphs, as ambiguous structure. Current methods usually consider single graph settings [12, 42], focusing on predicting future edges between two nodes solely based on their local structure. Therefore, these methods struggle to effectively model ambiguous structures under cross-domain setting. This limitation not only impedes the model's ability to accurately learn the meaning of ambiguous structures in multiple graphs but also hinders its capability to infer future edges correctly in target graphs, especially when the graphs contain many ambiguous structures.

Differing from current methods that are limited to local structure, this paper delves into cross-domain link prediction from a perspective of continual evolution [2, 18]. Intuitively, within the same graphs, their current evolution rule tends to consistent with their previous evolution pattern [2]. Therefore, if a model can input its historical evolution process, extract the implicit evolution pattern, and utilize it to infer future structure, naturally, this model can excel in cross-domain link prediction and understand ambiguous structures. As illustrated in Figure 1, solely considering the local structure, a model cannot predict edges between the red and blue nodes. However, if the model can capture the historical evolution

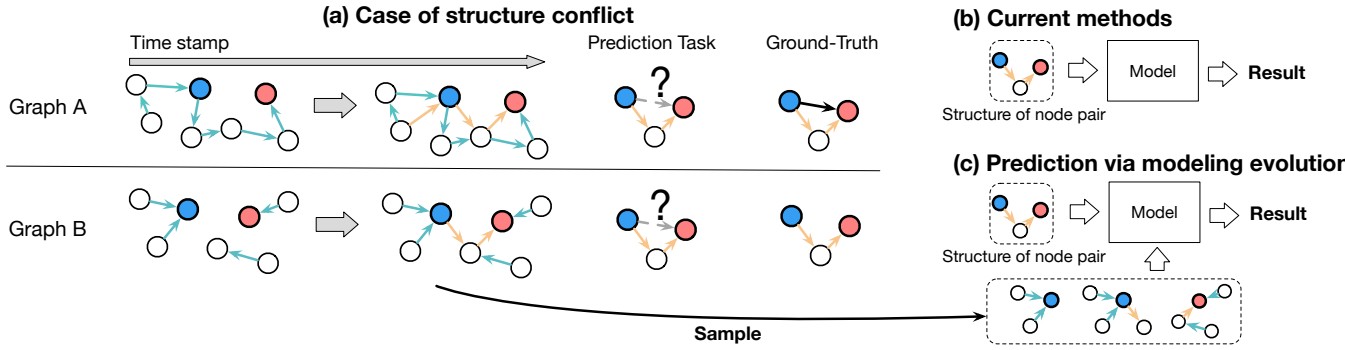

**Figure 1: (a) shows a case of structure conflict. Graph A follow a triadic closure process, while Graph B exhibits a contrasting process. (b) shows current methods cannot address this conflict. and (c) shows how prediction via modeling evolution, and it can address structure conflict.**

process of the corresponding graph and comprehend the associated patterns, it can discern the meaning of the input structure and make accurate predictions.

In line with this perspective, we propose CrossLink, a novel framework designed to enhance cross-domain link prediction by modeling evolutionary processes. CrossLink incorporates a *conditional link generation task* to integrate evolutionary modeling with link prediction, allowing for evolution-specific link prediction that effectively addresses structural ambiguity in cross-domain scenarios. More specifically, CrossLink interprets the graph evolution process as a continuous and concurrent link prediction task for all nodes. It represents this process through an evolution sequence that includes sampled link prediction tasks from the graph's history, denoted as $[\text{pair}_1, y_1^{t_1}, ...\text{pair}_K, y_K^{t_K}]$, where $\text{pair}_i$ denotes the $i$-th node pair, and $y_i^{t_i}$ indicates the emergence of an edge between the nodes of the pair after $t_i$. This sequence, along with the relationship between node pairs and their corresponding labels, reflects the underlying evolution pattern. When a target link prediction task is encountered, the framework integrates the new node pair into the evolution sequence, creating a merged sequence, represented as $[...\text{pair}_K, y_K^{t_K}, \text{pair}_{new}, ?]$. Consequently, evolution-specific link prediction becomes the task of predicting the next label in this sequence based on all prior elements, a process we term conditional link generation (CLG). To achieve CLG, CrossLink initially vectorizes the local structure of each node pair within the sequence using a structural component. It then employs a transformer-decoder architecture to predict the next edge label based on the entire sequence. Furthermore, CrossLink takes advantage of dynamic graph characteristics and includes careful engineering implementations [27] to enable parallel training and efficient inference.

We utilized six dynamic graphs from diverse domains, comprising a total of 6 million edges, to train CrossLink. Extensive experiments are conducted on eight untrained graphs to assess the effectiveness of CrossLink in cross-domain link prediction. When compared to five advanced dynamic graph models operating under cross-domain settings with the same training graphs, CrossLink demonstrated an 11.40% improvement in terms of link prediction *average precision (AP)* across the eight datasets. More remarkably, the cross-domain performance of CrossLink surpasses full-supervised performance of eight advanced baselines on six datasets. Detailed ablation studies indicate that CrossLink indeed improved by learning evolving patterns. Furthermore, analysis of training data showcases CrossLink's potential for scale-up to larger datasets and more parameters.

Our contributions are outlined as follows:

- Pioneering exploration of cross-domain link prediction, which is promising and can benefit broad applications.
- We propose a transformer-based model, CrossLink, which conducts cross-domain link prediction by integrating evolution process modeling, achieving state-of-the-art performance.
- Comprehensive experiments on 14 datasets and 8 advanced baselines, providing valuable insights to the community.

## 2 RELATED WORK

**Dynamic link prediction**. Dynamic graphs can model temporal interactions in various real-world scenarios [13], and link prediction is a crucial task with widespread applications [16]. Leveraging node representation for predicting edges has been widely employed in numerous applications [13]. Some studies focus on learning dynamically representing nodes based on self-supervised methods [11, 44]. With the advent of Graph Neural Networks (GNNs), some research endeavors to tackle link prediction through end-to-end learning. However, since GNNs are primarily designed for static graphs [10, 35], various approaches have been proposed to incorporate time-aware structures with GNNs, such as time encoders [40] and sequence models [17, 25, 31] More recently, some studies have proposed link prediction by employing neighbor sampling and unifying the representation using sequential models [37, 42] to achieve state-of-the-art performance on dynamic graphs [12]. However, these methods are often limited to single datasets. Currently, the exploration of cross-domain link prediction still stays in statistic [32] or static graphs [5, 9]. Therefore, this paper concentrates on cross-domain link prediction on dynamic graphs, aiming to benefit diverse real-world applications.

**Foundation model**. Currently, foundation models have been widely researched in various fields. For example, BERT [8], GPT-2/3

[4, 30], and LLAMA [33] in natural language processing (NLP); CLIP [29], and SAM [14] in CV. These methods have gained significant success in various applications. Since their pre-training tasks naturally support task-specific descriptions [3] (e.g., prompts [39]), one trained model can be applied to a variety of downstream tasks [20]. However, the exploration of foundational models in graphs is confined solely to static graphs [1, 19]. These inspire us to build a cross-domain link prediction model for dynamic graphs, where task-specific descriptions are typically the evolution of the graph.

## 3 BACKGROUND

**Dynamic graph**: This paper considers a scenario that has numerous graphs for diverse domains. We represent domains as $\mathcal{D} = \{0, 1, ...\}$. Each domain $d \in \mathcal{D}$ corresponds to a dynamic graph, expressed as a sequence of evolving links denoted by $\mathcal{E}_d = \{(e_1, t_1), (e_2, t_2), ...\}$. Here, $e_i$ signifies a dyadic event occurring between two nodes $u_i, v_i$. We also assume that each node $u_i$ exclusively belongs to a specific domain $d_i$, represented as $d_i = \phi(u_i)$. For a node $u_i \in \mathcal{E}_d$, $d_i = d$.

**Link Prediction on $\mathcal{E}_d$**: the task of dynamic link prediction is defined as using the previous edge set $\mathcal{E}_d^t = \{(e_i, t_i) \mid t_i \leq t\}$ to predict the future edges $\mathcal{E}'_d^t = \{(e_i, t_i) \mid t_i > t\}$.

From a machine learning perspective, the link prediction model can be defined as $\mathcal{F}(\mathcal{E}_d^t)$, where $\mathcal{E}_d^t$ serves as the input for the model. Most past works focus on single domain scenarios, which means the $\mathcal{F}(\cdot)$ is also trained by $\mathcal{E}_d^t$, i.e., *End2End training*. However, this paper focuses on a more challenging task: **cross-domain link prediction**, which problem can be formalized as follows:

**Problem definition**: Consider two sets of domains: $\mathcal{D}_{train}$ and $\mathcal{D}_{infer}$. The objective of this paper is to train one model, denoted as $\tilde{\mathcal{F}}(\cdot)$, utilizing multi-graphs from $\mathcal{D}_{train}$. For each untrained dynamic graph from $d \in \mathcal{D}_{infer}$ at timestamp $t$, $\tilde{\mathcal{F}}(\cdot)$ can directly predict $\mathcal{E}'_d^t$ based on $\mathcal{E}_d^t$ without the need for any further training process.

## 4 PROPOSED METHOD: CROSSLINK

### 4.1 Overview

This paper centers on proposing a model that is capable of cross-domain link prediction. Intuitively, diverse graphs encompass various physical meanings in cross-domain scenarios, which are harmful for training on numerous graphs and cross-domain generalizing. Hence, we introduce CrossLink, designed for training and inference on diverse datasets. The core idea of CrossLink is straightforward: it conducts link prediction based on the evolving patterns of the corresponding graph. CrossLink trained by **conditional link generation (CLG)** that can integrate the modeling of the evolution and link prediction and using caching.

**Outline**: When conducting a specific link prediction on a given graph, the CrossLink framework unfolds as follows: (1) Depict the graph's evolution process via a sequence of link prediction tasks with ground-truths; (2) Perform evolution-specific link prediction based on both nodes' representations and the evolution process. Then in the training stage, we merge (1) and (2) as conditioned link generation for parallel training. And in inference, we adapt caching (1) by KV-caching to accelerate (2).

## 4.2 Represent Graph Evolution by Sequence

We next show the details of how CrossLink represents the evolution process of a graph by a sequence of link prediction tasks. The evolution of a graph involves an ongoing process of sequential link generation. Naturally, graphs in different domains exhibit distinct generation sequences. Therefore, CrossLink directly models the historic generation sequences of the graphs to learn the evolution pattern.

Consider a dynamic graph $\mathcal{E}_d^t$ in the domain $d \in \mathcal{D}$ at timestamp $t$. CrossLink can sample a series of node pairs from this graph. Formally, the sampled edge pairs are represented as:

$$\tilde{\mathcal{E}}_{d_K}^{t_K} = \{...(u_i, v_i, t_i, y_{u_i,v_i}^{t_i})...(u_K, v_K, t_K, y_{u_K,v_K}^{t_K})\} \quad (1)$$

where $t_i < t_{i+1} < t_K < t$, $d_i = d_K = d$, and $y_{u_i,v_i}^{t_i} \in \{0, 1\}$ denotes whether $u_i$ is connected to $v_i$ at timestamp $t_i$. This sequence of pairs is a subset of the whole link generation process of $\mathcal{E}_d^t$, which reflects the intrinsic evolution pattern of the graph in domain $d$.

The evolution pattern especially is the correlation between the historical structure of two nodes and their future edge. Therefore, CrossLink uses a graph encoder to obtain the representation of all nodes in $\tilde{\mathcal{E}}_{d_K}^{t_K}$. Formally, for a pair $(u_i, v_i, t_i, y_{u_i,v_i}^{t_i})$, we derive the representations of $u_i$ and $v_i$ at timestamp $t_i$:

$$Z_{u_i}^{t_i}, Z_{v_i}^{t_i} = \textbf{GraphEncoder}(\mathcal{G}_{u_i}^{t_i}, \mathcal{G}_{v_i}^{t_i}, \mathcal{T}(t_*)) \quad (2)$$

Where $\mathcal{G}_{u_i}^{t_i}$ is the local subgraph of $u_i$ before $t_i$. Since the primary focus of this paper is not on representing the dynamic graph structure, we use **GraphEncoder**$(\cdot)$ as a general notation for a dynamic graph representation method, see more details in Appendix A.1. In this paper, we employ DyGFormer [42] as the instance of the graph encoder. Then, we can obtain the representation of $\tilde{\mathcal{E}}_{d_K}^{t_K}$:

$$\mathcal{P}(\tilde{\mathcal{E}}_d^{t_K}) = \{(Z_{u_0}^{t_0}, Z_{v_0}^{t_0}, y_{u_0,v_0}^{t_0})...(Z_{u_K}^{t_K}, Z_{v_K}^{t_K}, y_{u_K,v_K}^{t_K})\} \quad (3)$$

Here, $\mathcal{P}(\tilde{\mathcal{E}}_d^{t_K})$ is denoted as the **evolving sequence** in this paper, which contains rich information about $\mathcal{E}_d^t$. On one hand, the structure distribution of graph of $\mathcal{E}_d^t$ can be described by $Z_{u_i}^{t_i}$. On the other hand, the evolving pattern can be reflected by the correlation between node representation and corresponding $y_{u_*,v_*}^{t_*}$.

### 4.3 Link Prediction based on Graph Evolution

Next, CrossLink simultaneously models the gained evolving sequence and the representation of the target node to perform evolution-specific link prediction via a transformer.

Let $u_{K+1}, v_{K+1}, t_{K+1}$ denote an edge in domain $d$ to be predicted, where $t_{K+1} > t$. According to our assumption, the predicted $y_{u_{K+1},v_{K+1}}^{t_{K+1}}$ is not only related to the temporal structure of $u_{K+1}$ and $v_{K+1}$ but also to the evolution pattern. To gain a better understanding of the structure of $u_{K+1}$ and $v_{K+1}$, CrossLink employs the same graph encoder **GraphEncoder**$(\cdot)$ as in Eq. 2 to model the representation of nodes $u_{K+1}$ and $v_{K+1}$ before timestamp $t_K$, denoted as $Z_{u_{K+1}}^{t_{K+1}}$ and $Z_{v_{K+1}}^{t_{K+1}}$. Therefore, $Z_{u_{K+1}}^{t_{K+1}}$ exists in the same representation space as nodes in the sampled evolution pattern, making it easier to model their correlation.

Next, CrossLink integrates the representation of the predicting node pair with the graph evolution pattern represented by $\mathcal{P}(\tilde{\mathcal{E}}_d^{t_K})$. Given that evolution patterns are described in sequence form, for

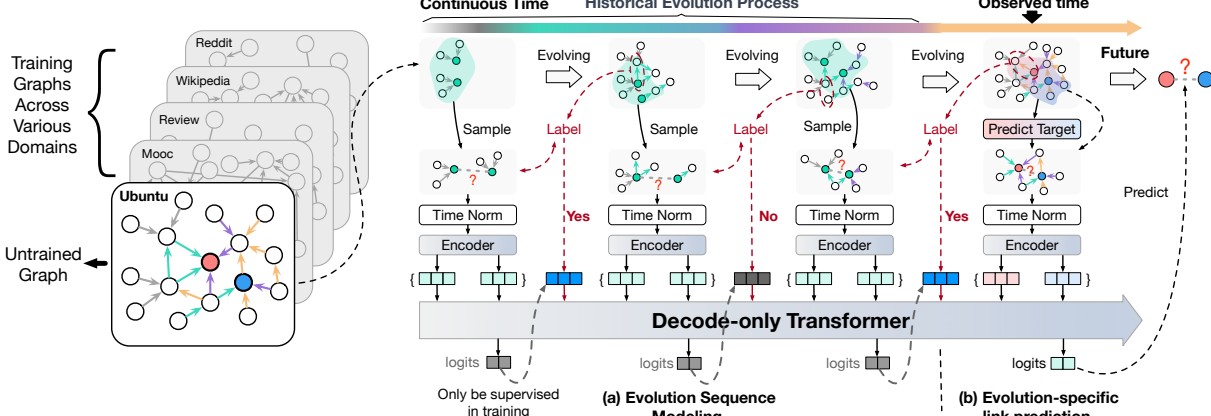

**Figure 2: Framework of CrossLink. (a) Models the graph's evolution process via a sequence of link prediction tasks with ground truths; (b) Evolution-specific link prediction based on both nodes' representations and the evolution process.**

consistency, the predicting node pair is also treated as a sequence, facilitating their combination through sequence concatenation. Subsequently, CrossLink employs a Transformer to sequentially model them, formalized as:

$$\{...H_{v_{K+1}}\} = \textbf{Transformer}([\mathcal{P}(\tilde{\mathcal{E}}_d^{t_K})||Z_{u_{K+1}}^{t_{K+1}}, Z_{v_{K+1}}^{t_{K+1}}]) \quad (4)$$

This is a 12-layer decode-only Transformer (with causality masking), adopted by many foundation models such as GPT-2 [30]. Its sequential modeling aligns with the graph evolving process as well as our target, namely link prediction based on evolution. Therefore, we use the last hidden state $H_{v_{K+1}}$ to predict the next token, as $H_{v_{K+1}}$ contains all input information and the final prediction result formalized as $y'^{t_{K+1}}_{u_{K+1},v_{K+1}} = f_\theta(H_{v_{K+1}})$, it optimized by cross-entropy with the ground truth, i.e., $\textbf{CE}(y'^{t_{K+1}}_{u_{K+1},v_{K+1}}, y^{t_{K+1}}_{u_{K+1},v_{K+1}})$.

### 4.4 Training and Inferring

**Parallel training by conditioned link generation**: Revisiting the representation used to model the evolving $\mathcal{P}(\tilde{\mathcal{E}}_d^{t_K})$, where each sampled pair is ordered by timestamp, i.e., in

$\{...(Z_{u_{i-1}}^{t_{i-1}}, Z_{v_{i-1}}^{t_{i-1}}, y_{u_K,vi-1}), (Z_{u_i}^{t_i}, Z_{v_i}^{t_i}, y_{u_K,v_i}^{t_i})...\}, t_{i-1} < t_i$.

Additionally, since $Z_{u_i}^{t_i}$ and $Z_{v_i}^{t_i}$ represent the structure before $t_i$ if we use $Z_{u_i}^{t_i}$ and $Z_{v_i}^{t_i}$ to predict $y_{u_i,v_i}^{t_i}$, this process can be treated as a link prediction task based on a shortened evolving pattern. Formally, let $\mathcal{P}(\tilde{\mathcal{E}}_d^{t_{i-1}}) \subset \mathcal{P}(\tilde{\mathcal{E}}_d^{t_K})$ denote all sampled node pairs before $t_i$. This link prediction task can be expressed as:

$$\{...H_{v_i}...\} = \textbf{Transformer}([\mathcal{P}(\mathcal{E}_d^{t_{i-1}})||Z_{u_i}^{t_i}, Z_{v_i}^{t_i}||...]) \quad (5)$$

Since CrossLink employs a decoder-only transformer, the sequence following of $Z_{v_i}^{t_i}$ does not influence the result of $H_{v_i}$. Consequently, based on $H_{v_i}$, we can reuse the prediction head to make a link prediction of the following content of $t_i$, denoted as $y'^{t_i}_{u_i,v_i} = f_\theta(H_{v_i})$.

Optimizing $y'^{t_i}_{u_i,v_i}$ offers several advantages. On the one hand, it can enhance the robustness of link prediction with evolution, as it essentially optimizes the prediction results based on evolving sequences of different lengths and combinations. On the other hand, it also can enhance the modeling evolution. Link prediction

is modeling the evolution pattern of dynamic graphs. Therefore, the intermediate result of link prediction can reflect the characteristics of evolution patterns. Meanwhile, the modeling of evolution patterns is optimized by successor link prediction which is based on evolution patterns. Therefore, this optimization is typically a **multi-domain training** model for better-comprehending evolution patterns.

Therefore, considering one sample in domain $d$, the loss of CrossLink:

$$\mathcal{L}_d(u_{K+1}, v_{K+1}, t_{K+1}) = \sum_{i=1}^{K+1} \textbf{CE}(y'^{t_i}_{u_i,v_i}, y^{t_i}_{u_i,v_i}) \quad (6)$$

Where $d_i = d_{K+1} = d$, and $K + 1$ is the max evolving sequence number. Since the last predicting pair also has ground truth, so all the pairs essentially generate the next ground-truth current nodes' representations and previous involution process. Thus, we term the entire process of Eq (6) as **conditioned link generation (CLG)**.

Finally, considering the multi-domain setting, the whole loss of CrossLink is: $\mathcal{L}_{all} = \sum_{d=0}^{\mathcal{D}_{train}} \mathcal{L}_d(*)$. It is worth noting that different domains would not be cross-sampled. Each computational process of $\mathcal{L}_d(*)$ occurred within a single graph in domain $d$. Algorithm 1 in the Appendix shows more details. **Time augmentation**: To minimize the disparity in the physical meaning of different graphs, we conduct time normalization for each subgraph $\mathcal{G}_{u_i}^{t_i}$ used in Eq (2). In $\mathcal{G}_{u_i}^{t_i}$, each time $t_{u,*}$ is normalized by $\tilde{t}_{u,*} = \frac{\alpha(t_i - t_{u,*})}{t_i}$, where the $\alpha$ is super-parameters. Additionally, to enhance the models' robustness to the temporal aspects and improve their ability to learn from evolving processes in link prediction, we introduce a time shuffle. For all times in one sequence, the normalized time is shuffled using two random seeds, $\hat{t}_{u,*} = \beta(\tilde{t}_{u,*} + \gamma)$. Since the random seeds are consistent for one sequence, CrossLink comprehends the temporal meaning by modeling the context of the sequence.

**Efficient inference**: In inference on a downstream graph, the construction of $\tilde{\mathcal{E}}_d^{t_K}$ in Eq (4) occurs only once. Subsequently, it can be reused for all link predictions. All computation results related to $\tilde{\mathcal{E}}_d^{t_K}$, including intermediate results in the decode-only transformer,

**Table 1: Performance of various methods regarding cross-domain link prediction. We report their *Average Precision* (average of 3 runs and omit by %) across eight graphs. Methods above the double horizontal lines present adopt cross-domain settings, and "*SOTA (End2End)*" denotes the best performance of eight baselines in End2End training (See details in Table 7). Bold and underline indicate the best and the second best performance respectively, and $\pm$ represents the variance. All subsequent tables utilize the same notations and metrics.**

| | Enron | UCI | Nearby | Myket | UN Trade | Ubuntu | Mathover. | College |
|---|---|---|---|---|---|---|---|---|
| Random | 54.85 ±9.26 | 55.17 ±9.14 | 52.22 ±9.38 | 53.06 ±7.05 | 50.43 ±6.21 | 53.61 ±5.98 | 52.47 ±9.92 | 55.17 ±9.26 |
| TGAT | 61.23 ±0.91 | 82.88 ±0.11 | 71.50 ±0.36 | 66.80 ±0.85 | 51.43 ±1.73 | 62.51 ±0.24 | 72.00 ±0.18 | 82.88 ±0.11 |
| CAWN | 81.00 ±1.05 | 94.16 ±0.01 | 69.10 ±0.34 | 71.46 ±0.17 | 56.20 ±0.58 | 58.18 ±0.53 | 68.23 ±0.37 | 94.16 ±0.01 |
| GraphMixer | 55.84 ±2.26 | 83.66 ±0.04 | 74.25 ±0.25 | 59.24 ±2.35 | 54.69 ±0.03 | 71.28 ±0.99 | 79.21 ±0.66 | 83.66 ±0.04 |
| TCL | 75.84 ±0.37 | 87.08 ±0.28 | 69.96 ±0.03 | 67.73 ±0.40 | 54.26 ±0.27 | 66.26 ±0.19 | 74.31 ±0.16 | 87.08 ±0.28 |
| DyGFormer | 90.80 ±0.54 | 93.86 ±0.19 | 71.74 ±0.39 | 71.34 ±0.46 | 55.17 ±2.37 | 65.95 ±0.82 | 75.70 ±0.34 | 93.86 ±0.19 |
| **CrossLink** | **91.60** ±0.47 | **96.02** ±0.07 | **89.61** ±0.92 | **87.72** ±0.11 | **60.52** ±0.29 | **87.44** ±0.76 | **89.13** ±0.33 | **96.04** ±0.07 |
| *SOTA (End2End)* | *92.47* ±0.12 | *95.79* ±0.17 | *89.32* ±0.05 | *86.77* ±0.00 | *66.92* ±0.07 | *84.82* ±0.03 | *89.04* ±0.08 | *94.75* ±0.03 |

can be cached. Thus, the time complexity during inference is only $O((K + 1)C)$ for each prediction, based on caching. Here, $K + 1$ represents the sequence length in self-attention, and $C$ accounts for all other matrix operations. With the incorporation of engineering optimizations, CrossLink, once trained, can effortlessly conduct efficient inference on very large-scale graphs.

## 5 EXPERIMENT

### 5.1 Experimental Settings

To assess the efficacy of CrossLink in cross-domain link prediction, we conduct an experiment involving 14 datasets and 9 baselines under 2 settings.

**Datasets**: Referencing prior research and current benchmarks [12, 42], this paper selects 14 unique and distinct graphs. As Table 3 shows, each link prediction task can be associated with a *specific real-world application*. Six graphs, namely Mooc, Wikipedia (abbreviated as Wiki), LastFM, Review, UN Vote, and Reddit, are randomly chosen for training, collectively comprising eight million dynamic edges. The remaining eight graphs used for zero-shot inference include Enron, UCI, Nearby, Myket, UN Trade, Ubuntu, Mathoverflow (abbreviated as Mathover.), and College. Due to the varying features of nodes across different datasets, we standardize the node and edge features of all data to be 0. Further details can be found in Appendix C.1.

**Baselines**: We select eight state-of-the-art (SOTA) dynamic graph models as our baselines, including TGAT [40], TGN [31], DyRep[34], Jodie[17], CAWN[37], GraphMixer[7], TCL[36], and DyGFormer[42]. It's important to note that TGN, DyRep, and Jodie are exclusively utilized under the End2End setting, as their designs are not suitable for cross-domain settings. For additional information, please refer to the Appendix C.2.

**Training setting and evaluation metrics**: This experiment mainly involves 2 training settings. (1) Cross-domain: all models are trained using the same set of six training graphs and evaluated on the other eight evaluated datasets. (2) End2End: independently train on the train-set and early stopping based on the validation set of 8 evaluated graphs. To enable a comparative analysis of cross-domain link prediction with End2End training, we chronologically

split each evaluation dataset into training, validation, and testing sets at a ratio of 70%/15%/15%. Additionally, we employ random negative sampling strategies for evaluation, a common practice in previous research. We report different performances of different models under different settings on the test-set sets. Due to the inclusion of extensive datasets and experiments, all the reported results are the *Average Precision (AP)* on the test set.

**Implementation details**: Our experiment is conducted based on DyGLib [42], see Appendix C.2 for more details.

### 5.2 Experiment Result

To assess CrossLink's effectiveness in cross-domain link prediction, we conduct a comparison with baselines in a cross-domain setting. In this setting, CrossLink and the baselines are trained by six graphs and subsequently evaluated on other eight untrained graphs. We further compare the cross-domain performance with full-supervised baselines (End2End), where baselines are trained by the train-set of each evaluated graph. Importantly, both settings utilized the same test sets. The reported results showcase the best End2End performance across the 8 baselines. Table 1 reveals three key insights from the experiment results:

*Insight 1. Link prediction is a generalized task across different domains*: (1) Existing baselines naturally demonstrate cross-domain transferability in link prediction. Compared to random prediction, all methods exhibit a 40.33% improvement after training on alternative datasets. Enron, UCI, and College stand out as easily transferable datasets, with the best performance of methods under cross-domain settings on these datasets being only 0.21% lower than the performance of the End2End setting on average. These results suggest that link prediction is a task that can be generalized across diverse domains. (2) However, current methods show instability under cross-domain settings. For example, GraphMixer performs second-best in Nearby and Mathoverflow but struggles in Enron, UCI, and Myket.

*Insight 2. CrossLink gains comprehensive improvement on cross-domain link prediction*: CrossLink differs from baselines that rely solely on the temporal structure of nodes for predicting links. Instead, CrossLink takes into account both the inherent

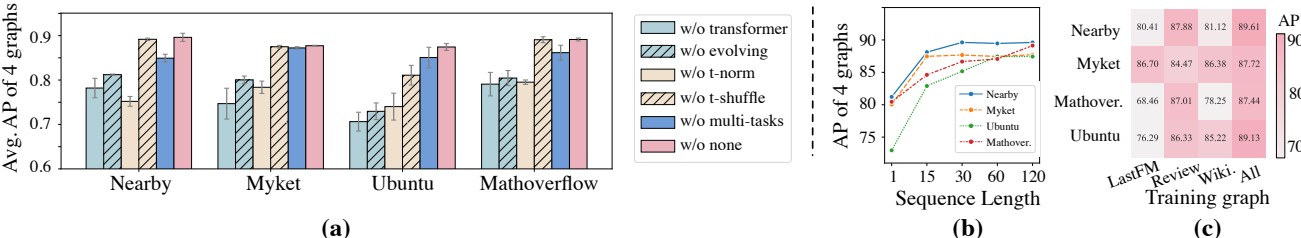

**Figure 3: Analysis result of CrossLink regarding multi-domain training. (a) shows the result of ablation studies, where "w/o" removes a certain component of our model. (b) shows the performance of CrossLink adopts different maximum sequence lengths (both training and inference). (c) indicates the performance on evaluated graphs that model solely trained by a specific graph. See more detailed results in Table 8, Table 9, and Table 10, respectively.**

patterns within predicting graphs and the temporal structure of nodes in link prediction. The experiment results demonstrate that CrossLink achieves significant improvement across various graphs. Compared to the best performance of all baselines under the same training setting, CrossLink showcases a remarkable improvement of over 11.40% (on an average of eight graphs). Particularly on datasets Nearby, Myket, Ubuntu, and Mathoverflow, the average improvement reaches up to 19.66% on average. These results thoroughly illustrate the advancements of CrossLink in cross-domain link prediction.

**Insight 3. CrossLink surpasses fully supervised performance on some graphs**: Furthermore, we compare CrossLink with the best performance of 8 baselines under End2End (SOTA of End2End). Surprisingly, on these 8 datasets, CrossLink outperforms the SOTA of End2End on 6 datasets, particularly on Ubuntu, with an impressive improvement of nearly 13.69%. For the datasets where CrossLink does not surpass, CrossLink is only 5.25% lower than End2End on average. This exciting result highlights the potential of CrossLink. Furthermore, it validates the feasibility of cross-domain link prediction.

Besides, we also observe the performance of five baselines and CrossLink on the test sets of six training datasets, and it also gains the SOTA performance on them. Besides, we find that training on more datasets indeed leads to poorer performance for some baselines, but CrossLink does not exhibit the same phenomenon. See more details in Table 4.

## 5.3 Analysis on Multi-domain Training

To delve deeper into why CrossLink performs well, we conduct a comprehensive analysis of its training process. To present the results more clearly, all observations are based on the four most improved datasets, namely Nearby, Myket, Ubuntu, and Mathoverflow. We observe the impact of the model's different components through ablation studies and then explore the effects of training data and model size.

**Insight 4. All components of CrossLink are important**: We conducted ablation studies to observe the influence of each component of CrossLink. According to the results shown in Figure 3, we have gathered several observations as follows:
- Modeling evolution drives CrossLink's performance improvement. When we exclude the evolving pattern (denoted as "w/o evolving"), CrossLink showed an average decrease of 11.11%

across four datasets. This outcome suggests that capturing the evolution process is the key to CrossLink's advancement.
- Multi-task training indeed enhances CrossLink. Multi-task training aids CrossLink in better modeling the evolution sequence. Removing the evolution-based multi-task learning led to an average AP decrease of 2.96% across these four datasets.
- Maximum evolution length can influence model performance: we further explore the impact of the maximum evolution length on the model (i.e., $K + 1$ in Eq (4)). As observed, with a rise in the evolution sequence length, the model's average ranking gradually increases.
- Time normalization is also crucial. Eliminating time normalization (denoted as "w/o t-norm") results in an average decrease of 13.21% across these datasets. This indicates that although evolution is crucial, it still requires a time normalization that effectively operates on a unified scale. Additionally, appropriately perturbing the normalized time can aid the model in more effective transfer. The removal of random shuffling ("w/o t-shuffle") also resulted in a decrease of 2.02%.

**Insight 5. Multi-domain training can prompt the model's generalization**: We also compare CrossLink with training on single dataset scenarios. As illustrated in Figure 3 (c), training on single datasets exhibits unstable performance across four datasets. In contrast, training on multi-datasets maintains stability on these four datasets and outperforms the best performance achieved by training on single datasets for each evaluated dataset.

**Insight 6. Large-scale and diversity of training datasets and suitable model size also important for CrossLink**:
- The model's performance increases with the expansion of the dataset scale. We train CrossLink under different training sizes (measured by number of edges), which are uniformly sampled from six training datasets. As shown in Figure 4 (a), with the increase in the number of training samples, the model's performance across four datasets gradually rises. Besides, we also observe changes in the model under different numbers of training datasets. As shown in Figure 4 (b), as the increase of graphs' number, the model's performance exhibits overall stable growth.
- The best-hidden size of the model grows with the increase of datasets. We initially show the performance of models with different hidden sizes under 6M training sets. As Figure 4 (c) shows, the best-hidden size of CrossLink is 128. We also examine the best-hidden size under other scales of training sets. As Figure 4 (d)

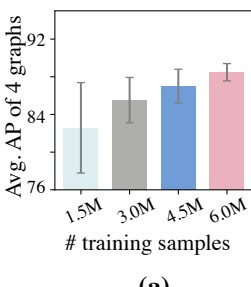
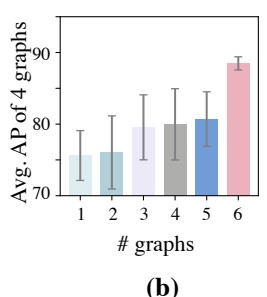
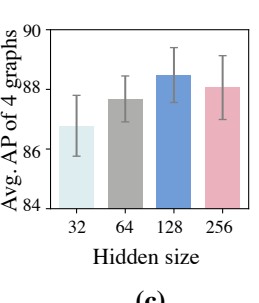
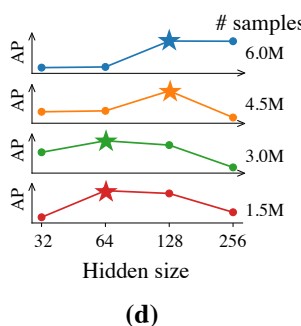

| (a) | (b) | (c) | (d) |

**Figure 4: Performance of CrossLink with diverse settings. (a) show the performance of the model improves with more training samples. (b) shows the performance of CrossLink is influenced by the number of training graphs. (c) shows the best-hidden size of the model under 6M training samples. (d) further shows the best-hidden size under different training samples. See more details in Appendix.**

shows as the increase of training datasets, the optimal parameters of the model also decrease from 64 to 128.

- **CrossLink exhibits the potential for scale-up.** Based on the above results, we observe that under the same parameters, model performance can grow with the scale-up of training data. Besides, as the dataset size increases, the optimal hidden size also gradually rises. This implies that if we continue to increase the scale of training data and incorporate a larger model, we can future improve the performance of cross-domain link prediction.

Additionally, we observed the impact of the input evolution sequence on CrossLink and gain **Insight 7** and **Insight 8**. For detailed information, please refer to Appendix E.

## 6 CONCULSION

This paper introduces CrossLink, which explicitly models the evolution process of graphs and conducts evolution-specific link predictions. Extensive experiments are conducted on eight untrained graphs, showcasing the effectiveness of CrossLink in cross-domain link prediction. Detailed analysis results reveal that CrossLink improved by model evolution and exhibits potential. The success of CrossLink demonstrates the feasibility and generalizability of modeling evolution. Besides, CrossLink's surpassing of fully supervised baselines and the analysis of its scalability both indicate the potential of cross-domain link prediction.

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

# A DYNAMIC GRAPH MODELS

## A.1 framework of dynamic graph

Currently, many dynamic graph models based on end2end learning gained advanced performance on link prediction tasks. These methods learn the representation of nodes' dynamic structure in $\mathcal{E}_t^d$ and pair-wised predicting future edges. That means their input consists of a node pair with a timestamp $\{u_i, v_i, t_i\}$, and they output the probability of a link appearing between $u_i$ and $v_i$ at timestamp $t_i$.

More specifically, these works propose various graph encoders to represent the historical temporal structure of nodes. Therefore, when considering a predictive edge pair $u_i, v_i, t_i$, the general framework of a graph encoder is outlined as follows: (1) First, sample local subgraphs ***before*** timestamp $t_i$ for these two nodes, denoted as $\mathcal{G}_{u_i}^{t_i} = \{(e_{u,0}, t_{u,0}), (e_{u,1}, t_{u,1}), \ldots\}$ and $\mathcal{G}_{v_i}^{t_i}$. (2) Obtain the initial node/edge features through feature construction and time features using a Time encoder. (3) Feed these features into either a sequence-based neural network or a graph neural network to obtain their respective representations.

For instance, in the sampling strategy of the subgraph $\mathcal{G}_{u_i}^{t_i}$, DyG-Former only considers one-hop neighbors [42], while CAWN [37] utilizes several sequences based on a time-aware random walk.

Since encoding the temporal structure is not the main focus of this paper, these processes can be briefly formalized as follows:

$$Z_{u_i}^{t_i}, Z_{v_i}^{t_i} = \mathbf{GraphEncoder}(\mathcal{G}_{u_i}^{t_i}, \mathcal{G}_{v_i}^{t_i}, \mathcal{T}(t_*)) \quad (7)$$

Here, $\mathcal{T}(t_*) \in \mathbb{R}^{*\times h}$ represents the process of encoding all the time $t_* \in \mathbb{R}^{*\times 1}$ in the subgraphs regarding two nodes, $Z_{u_i}^{t_i}$ and $Z_{v_i}^{t_i}$ are the final node representations of nodes $u_i$ and $v_i$ at time step $t_i$. Subsequently, dynamic models can make link predictions based on them, denoted as $y'^{t_i}_{u_i,v_i} = f(Z_{u_i}^{t_i}, Z_{v_i}^{t_i})$, where the function $f(\cdot)$ typically employs MLPs or dot-product in previous works [7].

## A.2 DyGFormer

Here are the details of DyGFormer. DyGFormer only sample nodes 1-hop neighbors. Step 1: Sample 1-hop neighbors of $u$ and $v$:

$\mathcal{N}_u^t = \{(w_{u,0}, t_{u,0}), (w_{u,1}, t_{u,1}), \ldots (w_{u,k}, t_{u,k})\}$

Step 2: Padding and Time Encoding:

$\mathbf{C}_\theta(w_0) = f_\theta(\mathbf{Count}(w_0, \mathcal{N}_u^{t_i})) + f_\theta(\mathbf{Count}(w_0, \mathcal{N}_v^{t_i}))$

$P_{u_i,k}^{t_i} = \mathbf{Concat}(x_{u,k}, \mathbf{T}_\theta(t_{u,k}), \mathbf{C}_\theta(w_{u,k}))$

$X_{u_i}^{t_i} = \{P_{u_i,0}^{t_i}, P_{u_i,1}^{t_i} \ldots P_{u_i,K}^{t_i}\}$

$\tilde{X}_{u_i,v_i}^t = \mathbf{Patch}([X_{u_i}^{t_i} || X_{v_i}^{t_i}])$

Step 3: Encoding Transformer:

$Z_{u_i}^{t_i}, Z_{v_i}^{t_i} = \mathbf{Transformer}(\tilde{X}_{u_i,v_i}^t)$

## B  ALGORITHM OF CROSSLINK

Here is the details algorithm of CrossLink, note that SortInSeqBy-Time implies grouping according to SeqNum, sorting only within groups based on time. This randomization and grouping are primarily aimed at ensuring training variability and regularity in negative sample collection, thereby better maintaining consistency with inference scenarios.

---

**Algorithm 1** Multi-domain training of CrossLink

---

**Input:** $\mathbb{E} = \{\mathcal{E}_0, ..., \mathcal{E}_{|\mathcal{D}|}\}$, $m$
**for** $epoch$ in Range($maxEpoch$) **do**
   $TensorAllData = []$
   $GroundTruth = []$
   **for** $\mathcal{E}_i$ in $\mathbb{E}$ **do**
     $batchSingleData = $ Sample($\mathcal{E}_i$)
     **for** $(u, v, t)$ in $batchSingleData$ **do**
       $batchSingleData = $ SortByEdgeTime($batchSingleData$)
       $\tilde{v} = $ NegtiveSampe($u, v, \mathcal{E}_i, t$)
       $\mathcal{N}_u, \mathcal{N}_v, \mathcal{N}_{\tilde{v}} = $ NeighborSample($u, v, \tilde{v}, \mathcal{E}_i, t$)
       $\mathbf{C}(u, v) = $ CooNeighbor($\mathcal{N}_u, \mathcal{N}_v$)
       $\mathbf{C}(u, \tilde{v}) = $ CooNeighbor($\mathcal{N}_u, \mathcal{N}_{\tilde{v}}$)
       $TensorBatchData$.add([$\mathcal{N}_u, \mathcal{N}_v, \mathbf{C}(u, v), PosLabel$])
       $TensorBatchData$.add([$\mathcal{N}_u, \mathcal{N}_{\tilde{v}}, \mathbf{C}(u, \tilde{v}), NegLabel$])
       $BatchGroundTruth$.add($PosLabel$)
       $BatchGroundTruth$.add($NegtiveSampe$)
     **end for**
     $randomIndex = $ random()
     $BatchGroundTruth = BatchGroundTruth[randomIndex]$
     $TensorBatchData = TensorBatchData[randomIndex]$
     $BatchGroundTruth = SortInSeqByTime(BatchGroundTruth, seqNum)$
   **end for**
   $\mathbb{S}, \mathbb{R}, \mathbb{L} = $ **GraphEncode**($TensorAllData$)
   $evolvingSeq = $ **Concat**($\mathbb{S}, \mathbb{R}, \mathbb{L}, dim = 1$).reshape($-1, seqNum * 3, hiddenSize$)
   $hiddenstates = $ **Transformer**($evolvingSeq$)
   $hiddenstates = hiddenstates.$reshape($-1, 3, hiddenSize$)
   $logitsNextLabel = hiddenstates[:, 1, :]$
   $\mathcal{L} = $ **CrossEntropy**($logitsNextLabel, GroundTruth$)
   $\mathcal{L}$.backward()
**end for**

---

## C  EXPERIMENT SETTING

### C.1  Datasets

We use 8 datasets collected by [28] in the experiments, which are publicly available in the website[1]:

- *Wikipedia* can be described as a bipartite interaction graph, which encompasses the modifications made to Wikipedia pages within over a month. In this graph, nodes are utilized to represent both users and pages, while the links between them signify instances of editing actions, accompanied by their respective timestamps. Furthermore, each link is associated with a 172-dimensional Linguistic Inquiry and Word Count (LIWC) feature [26]. Notably, this dataset also includes dynamic labels that serve to indicate whether users have been subjected to temporary editing bans.
- *Reddit* is a bipartite network that captures user interactions within subreddits over one month. In this network, users and subreddits serve as nodes, and the links represent timestamped posting requests. Additionally, each link is associated with a 172-dimensional LIWC feature, similar to that of Wikipedia. Furthermore, this dataset incorporates dynamic labels that indicate whether users have been prohibited from posting.
- *MOOC* refers to a bipartite interaction network within an online educational platform, wherein nodes represent students and course content units. Each link in this network corresponds to a student's interaction with a specific content unit and includes a 4-dimensional feature to capture relevant information.
- *LastFM* is a bipartite network that comprises data concerning the songs listened to by users over one month. In this network, users and songs are represented as nodes, and the links between them indicate the listening behaviors of users.
- *Enron* documents the email communications among employees of the ENRON energy corporation spanning a period of three years.
- *UCI* represents an online communication network, where nodes correspond to university students, and links represent messages posted by these students.
- *UN Trade* dataset encompasses the trade of food and agriculture products among 181 nations spanning more than 30 years. The weight associated with each link within this dataset quantifies the cumulative value of normalized agriculture imports or exports between two specific countries.
- *UN Vote* records roll-call votes in the United Nations General Assembly. When two nations both vote 'yes' on an item, the weight of the link connecting them is incremented by one.

And, we still used other datasets collected from different places:

- *Nearby*. This dataset contains all public posts and comments, and can be downloaded in website[2]
- *Ubuntu* collected by [24]. A temporal network of interactions on the stack exchange website Ask Ubuntu[3] and every edge represents that user $u$ answered user $v$'s question at time $t$.
- *Mathoverflow* collected by [24]. A temporal network of interactions on the stack exchange website Mathoverflow[4] and every edge represents that user $u$ answered user $v$'s question at time $t$.
- *Review*, as described in [12], comprises an Amazon product review network spanning the period from 1997 to 2018. In this network, users provide ratings for various electronic products on a scale from one to five. As a result, the network is weighted, with both users and products serving as nodes, and each edge representing a specific review from a user to a product at a specific timestamp. Only users who have submitted a minimum of ten reviews during the mentioned time interval are retained in the network. The primary task associated with this dataset is predicting which product a user will review at a given point in time.

---

[1]https://zenodo.org/record/7213796#.Y1cO6y8r30o

[2]https://www.kaggle.com/datasets/brianhamachek/nearby-social-network-all-posts
[3]https://askubuntu.com/
[4]https://mathoverflow.net/

**Table 2: Statistics of the datasets. Above the dotted line are the six pre-training datasets. #Nodes, #Links, #Src/#Dst, #N&L respectively refer to the number of nodes, the number of links, the ratio of unique source nodes to unique destination nodes within the dataset, and the node&link feature dimension(– means not having the feature). Bipartite and Directed indicate whether the graph is a bipartite graph or a directed graph (denoted as 1F5F8 for yes and 2613 for no). / means remaining unknown.**

| Datasets | Domains | #Nodes | #Links | #Src/#Dst | #N&L Feat | Bipartite | Directed | Duration | Unique Steps | Time Granularity |
|---|---|---|---|---|---|---|---|---|---|---|
| MOOC | Interaction | 7,144 | 411,749 | 72.65 | – & 4 | 1F5F8 | 1F5F8 | 17 months | 345,600 | Unix timestamps |
| LastFM | Interaction | 1,980 | 1,293,103 | 0.98 | – & – | 1F5F8 | 1F5F8 | 1 month | 1,283,614 | Unix timestamps |
| Review | Rating | 352,637 | 4,873,540 | 1.18 | – & – | 2613 | 1F5F8 | 21 years | 6,865 | Unix timestamps |
| UN Vote | Politics | 201 | 1,035,742 | 1.00 | – & 1 | 2613 | 1F5F8 | 72 years | 72 | years |
| Wikipedia | Social | 9,227 | 157,474 | 8.23 | – & 172 | 1F5F8 | 1F5F8 | 1 month | 152,757 | Unix timestamps |
| Reddit | Social | 10,984 | 672,447 | 10.16 | – & 172 | 1F5F8 | 1F5F8 | 1 month | 669,065 | Unix timestamps |
| Enron | Social | 184 | 125,235 | 0.98 | – & – | 2613 | 1F5F8 | 3 years | 22,632 | Unix timestamps |
| UCI | Social | 1,899 | 59,835 | 0.73 | – & – | 2613 | 1F5F8 | 196 days | 58,911 | Unix timestamps |
| Nearby | Social | 34,038 | 120,772 | 0.80 | – & – | 2613 | 1F5F8 | 21 days | 99,224 | Unix timestamps |
| Myket | Action | 17,988 | 694,121 | 1.25 | – & – | 1F5F8 | 1F5F8 | / | 693,774 | / |
| UN Trade | Economics | 255 | 507,497 | 1.00 | – & 1 | 2613 | 1F5F8 | 32 years | 32 | years |
| Ubuntu | Interaction | 137,517 | 280,102 | 0.75 | – & – | 2613 | 1F5F8 | 2613 days | 279,840 | days |
| Mathoverflow | Interaction | 21,688 | 90,489 | 0.65 | – & – | 2613 | 1F5F8 | 2350 days | 107,547 | days |
| College | Social | 1,899 | 59,835 | 0.73 | – & – | 2613 | 1F5F8 | 193 days | 58,911 | Unix timestamps |

**Table 3: The aligned real-world application by each link prediction. Above the dotted line are the six training datasets.**

| | Type | Real-world Application |
|---|---|---|
| MOOC | Course Selecting | Weather a course in a specific content unit selected by a student. |
| LastFM | Song listened | Weather a particular song was listened to by a user. |
| Review | Product Reviewed | Weather a product is reviewed in Amazon. |
| UN Vote | Voting | Weather a nation vote "yes" on an item in the United Nations General Assembly. |
| Wikipedia | Entry Editing | Weather a user edited a specific entry in Wikipedia. |
| Reddit | Social Interaction | Weather user's interactions within subreddits. |
| Enron | Email Communication | Weather Two employees in ENRON energy corporation communicate by email. |
| UCI | Online Communication | Weather messages posted by students in a university. |
| Nearby | Post Commented | Weather a public post in Nearby is commented. |
| Myket | App installed | Weather an Android application installations within the Myket Android application market. |
| UN Trade | International Trade | Weather is a cumulative value of normalized agriculture imports or exports between two specific countries. |
| Ubuntu | Question Answered | whether a user answered another user's question in Ask Ubuntu. |
| Mathoverflow | Question Answered | Weather a user answered another user's question in Mathoverflow. |
| College | Online Communication | Weather is a private message sent between students in the University of California, Irvine. |

- *Myket* collected by [21]. This dataset encompasses data regarding user interactions related to application installations within the Myket Android application market[5].
- *College* collected by [23]. This dataset consists of private messages exchanged within an online social network at the University of California, Irvine. An edge denoted as $(u, v, t)$ signifies that user $u$ sent a private message to the user $v$ at the timestamp $t$.

## C.2 Implementation details

Our experiment is conducted based on DyGLib [42], an open-source toolkit with standard training pipelines including diverse benchmark datasets and thorough baselines.

Due to the varying number of edges in each dataset, we apply duplicate sampling to datasets with fewer edges to ensure consistency in the number of edges for each dataset. Following the setting of [42], we employ a 1:1 negative sampling strategy for the edges of each dataset, with three random seeds in both the training and inference phases to ensure that each method is trained or tested with the same negative samples in each randomization.

Our method and baselines utilize the default setup of DyGLib [42], with hidden sizes of 128 for time, edge, and node, employing AdamW as the optimizer and linear warm-up. Given the cross-domain scenario and the presence of negative sampling in link prediction, we

---
[5] https://myket.ir/

**Table 4: Link Prediction AP for training on the single dataset, comparing with training on all 6 datasets.**

| Dataset | Train | TGAT | CAWN | Mixer. | Former. | Ours |
|---|---|---|---|---|---|---|
| **Mooc** | Single | 80.76 | 83.22 | 72.09 | 83.55 | **83.84** |
| | All | 80.67 | 82.76 | 67.10 | 82.99 | **83.38** |
| | *Impv.* | *-0.11%* | *-0.56%* | *-7.44%* | *-0.67%* | *-0.55%* |
| **Wiki** | Single | 95.58 | 99.20 | 95.99 | 99.36 | **99.39** |
| | All | 95.21 | 99.21 | 91.85 | 99.31 | **99.40** |
| | *Impv.* | *-0.39%* | *-0.06%* | *-4.51%* | *-0.05%* | *+0.01%* |
| **LastFM** | Single | 67.85 | 86.29 | 70.74 | 89.75 | **91.82** |
| | All | 66.22 | 86.29 | 58.51 | 89.73 | **91.84** |
| | *Impv.* | *-2.46%* | *+0.00%* | *-20.90%* | *-0.02%* | *+0.02%* |
| **Review** | Single | 72.46 | 73.26 | 89.28 | 85.59 | **91.54** |
| | All | 70.26 | 77.62 | 82.90 | 84.87 | **91.53** |
| | *Impv.* | *-3.13%* | *+5.62%* | *-7.70%* | *-0.85%* | *-0.01%* |
| **UN Vote** | Single | 50.59 | 53.76 | 50.96 | **56.71** | 55.86 |
| | All | 50.87 | 53.98 | 50.96 | **56.53** | 55.63 |
| | *Impv.* | *+0.55%* | *+0.41%* | *+0.00%* | *-0.32%* | *-0.41%* |
| **Reddit** | Single | 91.33 | 98.58 | 91.34 | 98.98 | **99.11** |
| | All | 91.57 | 98.49 | 88.12 | 98.93 | **99.11** |
| | *Impv.* | *+0.26%* | *-0.09%* | *-3.65%* | *-0.05%* | *+0.00%* |

**Table 5: Detailed Link prediction Average Precision (AP) on the unseen datasets across various inference sequence lengths for a model trained by 120 maximum evolving length. ± indicates variance, and the remaining notations in the table are consistent with those in Table 15.**

| Seq Length | 1 | 29 | 59 | 89 | 119 |
|---|---|---|---|---|---|
| Nearby | 0.8225±0.0065 | 0.8516±0.0033 | 0.8669±0.0015 | 0.8865±0.0133 | **0.8961**±0.0092 |
| Myket | 0.8190±0.0139 | 0.8747±0.0012 | 0.8750±0.0035 | 0.8771±0.0023 | **0.8772**±0.0011 |
| Ubuntu | 0.7191±0.0009 | 0.7907±0.0218 | 0.7879±0.0132 | 0.8492±0.0174 | **0.8744**±0.0076 |
| Mathoverflow | 0.8042±0.0185 | 0.8434±0.0152 | 0.8421±0.0046 | 0.8731±0.0189 | **0.8913**±0.0033 |

observed that the validation set of the training data does not necessarily correlate directly with the performance of the evaluated dataset. Considering that real-world scenarios often involve training on the entire dataset and emphasize the robustness of methods, all our models were trained for 50 epochs, and results were reported based on the last epoch. Each model underwent three training iterations with three different random seeds, ensuring consistency in the edges and negative samples seen by all models in each epoch.

For our method, the graph encoder employed DyGFomer with time augmentation. The transformer had 12 layers, 8 attention heads, and a hidden size of 128. All details, including activation functions, layer normalization, and GPT2 decoder, are consistent throughout.

## D EXPERIMENT RESULT

## E ANALYSIS ON EVOLUTION SEQUENCE

We further analyze the properties of CrossLinkin cross-domain inference. We first investigate the impact of the sequence length of the evolution sequence in the evaluated graph. Subsequently, we next explore the effects on the model if an inappropriate evolution sequence is employed.

**Table 6: Detailed ink prediction Average Precision (AP) on the unseen datasets while using no in-context (# Seq = 1) or using the evolving pattern of other domain with sequence length 120. ±indicates variance, and the remaining notations in the table are consistent with those in Table 16.**

| | Seq = 1 | Using other evolving pattern | | | |
| --- | --- | --- | --- | --- | --- |
| | | Mooc | LastFM | Review | Reddit |
| Nearby | 0.82255±0.0065 | 0.80755±0.0106 | 0.81995±0.0048 | **0.87465**±0.0082 | 0.84335±0.0072 |
| Myket | 0.81950±0.0139 | 0.79295±0.0093 | 0.86225±0.0034 | 0.86625±0.0029 | **0.87775**±0.0008 |
| Ubuntu | 0.71915±0.0009 | 0.75345±0.0109 | 0.74955±0.0128 | **0.88775**±0.0033 | 0.79675±0.0110 |
| Mathoverflow | 0.80425±0.0185 | 0.81795±0.0183 | 0.81155±0.0229 | **0.89095**±0.0053 | 0.83675±0.0064 |

**Table 7: Link Prediction Average Precision (AP) for End2End training and inferring.**

| | Enron | UCI | Nearby | Myket | UN Trade | Ubuntu | Mathoverflow | College |
| --- | --- | --- | --- | --- | --- | --- | --- | --- |
| TGAT | 71.12±0.97 | 79.63±0.70 | 83.20±1.10 | 76.32±1.03 | 55.80±1.15 | 76.00±0.12 | 75.18±0.03 | 85.09±0.32 |
| TGN | 86.53±1.11 | 92.34±1.04 | 76.62±4.62 | 85.72±0.88 | 66.84±1.13 | 50.47±2.39 | 62.53±1.04 | 88.37±0.27 |
| DyRep | 82.38±3.36 | 65.14±2.30 | 73.72±0.52 | 86.05±0.09 | 54.61±0.48 | 64.56±0.29 | 73.63±0.21 | 65.72±1.87 |
| Jodie | 84.77±0.30 | 89.43±1.09 | 80.19±0.40 | 86.30±0.47 | 58.11±0.61 | 67.25±0.70 | 77.43±0.09 | 77.08±0.48 |
| TCL | 79.70±0.71 | 89.57±1.63 | 82.13±0.43 | 67.52±1.75 | 54.65±0.08 | 71.93±0.86 | 74.83±0.21 | 87.03±0.30 |
| CAWN | 89.56±0.09 | 95.18±0.06 | 85.01±0.16 | 77.64±0.25 | 61.35±0.22 | 75.05±0.40 | 79.27±0.10 | 94.58±0.05 |
| GraphMixer | 82.25±0.16 | 93.25±0.57 | 89.32±0.03 | 86.77±0.00 | 54.64±0.03 | 84.82±0.03 | 89.04±0.08 | 92.44±0.03 |
| DyGFormer | 92.47±0.12 | 95.79±0.17 | 84.70±0.24 | 85.12±0.04 | 66.92±0.07 | 76.00±0.12 | 80.43±0.02 | 94.75±0.03 |

**Table 8: Ablation experiments for CrossLink. Each line indicates the performance for link prediction evaluated by AP when certain aspects of our model's design are removed.**

| | | Enron | UCI | Nearby | Myket | UN Trade | Ubuntu | Mathoverflow | College |
| --- | --- | --- | --- | --- | --- | --- | --- | --- | --- |
| Network Structure | w/o decoder | 90.09±0.39 | 95.89±0.24 | 78.20±2.19 | 74.69±3.47 | 58.43±0.48 | 70.64±2.12 | 79.08±2.65 | 95.89±0.29 |
| | w/o evolving | 90.84±0.72 | 96.08±0.07 | 81.21±0.14 | 80.03±0.86 | 54.72±0.49 | 72.96±1.88 | 80.45±1.70 | 96.09±0.07 |
| Time Processer | w/o t-norm | 85.86±0.50 | 95.47±0.17 | 75.20±1.11 | 78.37±1.39 | 58.68±1.19 | 74.02±3.03 | 79.52±0.50 | 95.47±0.17 |
| | w/o t-shuffle | 91.37±0.15 | 96.25±0.28 | 89.17±0.27 | 87.49±0.33 | 60.35±0.43 | 81.09±2.22 | 89.07±0.67 | 95.25±0.28 |
| Training Strategy | w/o multi-datasets | 91.08±0.50 | 95.57±0.08 | 87.88±0.43 | 86.70±0.77 | 57.52±0.88 | 86.33±0.89 | 87.01±0.30 | 95.57±0.08 |
| | w/o multi-tasks | 88.20±1.59 | 95.49±0.13 | 84.92±0.89 | 87.23±0.26 | 57.13±0.26 | 85.07±2.51 | 86.16±1.68 | 95.50±0.13 |
| | All Strategy | 91.60±0.47 | 96.02±0.07 | 89.61±0.92 | 87.72±0.11 | 60.52±0.29 | 87.44±0.76 | 89.13±0.33 | 96.04±0.07 |

**Table 9: The performance of link prediction in terms of Average Precision (AP) under various maximum training and inference sequence lengths is presented. The experiments were conducted on the same six datasets with consistent settings throughout. The values in the table represent the mean, and ± indicates the variance.**

| Maximum of Seq Length | 1 | 15 | 30 | 60 | 120 |
| --- | --- | --- | --- | --- | --- |
| Enron | 90.84±0.72 | 90.91±0.46 | 91.56±0.19 | 91.57±0.50 | 91.60±0.47 |
| UCI | 96.08±0.07 | 95.82±0.21 | 95.83±0.20 | 95.97±0.25 | 96.02±0.07 |
| Nearby | 81.21±0.14 | 88.12±0.29 | 89.62±0.29 | 89.46±0.22 | 89.61±0.92 |
| Myket | 80.03±0.86 | 87.47±0.20 | 87.67±0.39 | 87.45±0.45 | 87.72±0.11 |
| UNtrade | 54.72±0.49 | 58.35±0.47 | 58.99±0.86 | 60.07±0.54 | 60.52±0.29 |
| Ubuntu | 72.96±1.88 | 82.89±1.83 | 85.16±2.26 | 87.44±1.44 | 87.44±0.76 |
| Mathoverflow | 80.45±1.70 | 84.61±0.36 | 86.66±2.62 | 87.07±2.63 | 89.13±0.33 |
| College | 96.09±0.07 | 95.84±0.21 | 95.84±0.19 | 95.78±0.25 | 96.04±0.07 |

***Insight 7. Longer evolving on target dataset can improve the predicting performance***: During the inference process, we vary the length of the evolving pattern (denoted as seq-length). As shown in Table 15, the model's performance steadily increases with the length. This aligns with intuition, as a longer evolution sequence has more information so can better reflect the characteristics of a graph. This result also suggests that CrossLink's cross-domain link prediction capability is indeed attributed to the modeling of the evolution sequence on the target graph.

***Insight 8. We can also use other graphs to activate the model's performance.*** We further investigate how the model would

**Table 10: The performance of link prediction, as measured by the Average Precision (AP), was trained on different individual datasets and tested on another dataset. Different rows represent the datasets used for training, while different columns represent the datasets used for testing. The values in the table represent the mean, and ± indicates the variance.**

| | Mooc | LastFM | Review | UNvote | Wiki | Reddit |
| --- | --- | --- | --- | --- | --- | --- |
| enron | 82.46±1.91 | 90.19±1.11 | 60.14±4.96 | 71.01±3.76 | 91.08±0.50 | 86.58±1.68 |
| UCI | 93.02±0.25 | 95.57±0.08 | 85.12±2.72 | 79.56±0.83 | 95.26±0.16 | 95.02±0.25 |
| Nearby | 76.66±2.02 | 80.41±0.52 | 87.88±0.43 | 55.67±3.13 | 81.12±0.17 | 71.97±2.24 |
| Myket | 73.60±2.64 | 86.70±0.77 | 84.47±0.23 | 68.14±1.88 | 86.38±0.24 | 82.86±2.20 |
| UN Trade | 52.09±2.94 | 57.26±0.40 | 54.42±4.39 | 51.65±0.54 | 57.52±0.88 | 57.10±0.41 |
| Mathoverflow | 80.73±0.80 | 76.29±3.46 | 86.33±0.89 | 61.66±2.63 | 85.22±0.38 | 75.58±0.52 |
| Ubuntu | 71.46±1.92 | 68.46±2.59 | 87.01±0.30 | 55.63±2.68 | 78.25±0.44 | 69.01±0.53 |
| College | 93.01±0.25 | 95.57±0.08 | 85.12±2.72 | 79.56±0.83 | 95.26±0.16 | 95.03±0.24 |

**Table 11: The Average Precision (AP) of CrossLinktrained on a dataset of size 1.5 million for link prediction on an unseen dataset. Bold indicates the best results, underline indicates the second best results, and ± represents the variance.**

| hidden size | 32 | 64 | 128 | 256 |
| --- | --- | --- | --- | --- |
| Enron | 85.29±0.96 | 86.59±1.98 | 87.59±0.76 | 88.41±0.53 |
| UCI | 95.86±0.08 | 95.88±0.30 | 95.90±0.02 | 95.81±0.08 |
| Nearby | 83.84±0.25 | 86.21±0.76 | 85.62±0.51 | 84.62±1.17 |
| Myket | 86.75±0.22 | 87.51±0.29 | 87.39±0.45 | 86.98±0.39 |
| UN Trade | 55.21±1.77 | 55.51±0.48 | 56.62±0.88 | 55.97±0.23 |
| Ubuntu | 81.10±1.60 | 82.22±4.38 | 74.86±1.22 | 73.69±1.48 |
| Mathoverflow | 81.25±0.65 | 82.58±1.52 | 82.44±0.84 | 81.50±1.42 |
| College | 95.87±0.08 | 95.89±0.30 | 95.90±0.01 | 95.81±0.09 |

**Table 12: The Average Precision (AP) of CrossLinktrained on a dataset of size 3.0 million for link prediction on an unseen dataset. Bold indicates the best results, underline indicates the second best results, and ± represents the variance.**

| hidden size | 32 | 64 | 128 | 256 |
| --- | --- | --- | --- | --- |
| Enron | 90.15±0.25 | 90.75±0.52 | 90.67±0.92 | 90.76±0.61 |
| UCI | 96.10±0.18 | 96.16±0.05 | 96.16±0.05 | 95.94±0.15 |
| Nearby | 86.81±0.70 | 87.26±0.33 | 88.57±1.20 | 87.20±0.43 |
| Myket | 85.04±1.30 | 87.08±0.79 | 87.09±0.51 | 86.77±0.67 |
| UN Trade | 58.37±1.22 | 59.18±1.33 | 56.39±0.95 | 58.63±1.03 |
| Ubuntu | 83.32±1.50 | 83.91±0.40 | 82.55±2.41 | 79.55±0.73 |
| Mathoverflow | 83.58±1.45 | 84.06±0.29 | 83.88±0.37 | 82.95±1.15 |
| College | 96.11±0.18 | 96.17±0.05 | 95.90±0.01 | 95.84±0.14 |

behave if link prediction is based on evolving sequences from another graph. As shown in Table 16, the performance varies when models using evolving sequences from some graphs may lead to a decline in the model's performance on the target dataset, while others do not. On the one hand, this result suggests model indeed models the evolution sequence and makes evolution-specific predictions. On the one hand, it also implies a potential advantage of CrossLink: even when there are not enough edges on the target dataset, we also can use the evolution sequence of graphs similar to the target graph, thereby activating CrossLink's link prediction capability.

**Table 13: The Average Precision (AP) of CrossLinktrained on a dataset of size 4.5 million for link prediction on an unseen dataset. Bold indicates the best results, underline indicates the second best results, and ± represents the variance.**

| hidden size | 32 | 64 | 128 | 256 |
|---|---|---|---|---|
| Enron | 91.21±0.91 | 91.84±0.06 | 91.86±0.24 | 91.55±0.40 |
| UCI | 95.73±0.10 | 96.15±0.13 | 95.93±0.05 | 96.01±0.14 |
| Nearby | 87.71±0.13 | 87.96±0.49 | 89.49±0.29 | 88.65±0.47 |
| Myket | 86.17±0.43 | 87.39±0.49 | 87.52±0.47 | 86.77±1.33 |
| UN Trade | 57.10±2.59 | 59.48±0.93 | 58.49±1.54 | 59.16±0.88 |
| Ubuntu | 84.44±1.68 | 85.81±1.73 | 84.52±1.52 | 82.87±1.59 |
| Mathoverflow | 84.89±0.63 | 84.96±1.50 | 86.46±1.91 | 84.46±0.64 |
| College | 95.75±0.10 | 96.16±0.13 | 95.94±0.04 | 96.01±0.15 |

**Table 14: The Average Precision (AP) of CrossLinktrained on a dataset of size 6.0 million for link prediction on an unseen dataset. Bold indicates the best results, underline indicates the second best results, and ± represents the variance.**

| hidden size | 32 | 64 | 128 | 256 |
|---|---|---|---|---|
| Enron | 91.76±0.41 | 91.49±0.38 | 91.60±0.47 | 91.47±0.68 |
| UCI | 95.69±0.19 | 95.85±0.22 | 96.02±0.07 | 95.60±0.06 |
| Nearby | 88.03±0.40 | 88.90±0.72 | 89.61±0.92 | 89.16±0.78 |
| Myket | 87.12±0.33 | 87.68±0.07 | 87.72±0.11 | 87.01±0.66 |
| UN Trade | 57.33±2.06 | 59.64±0.65 | 60.52±0.29 | 60.58±0.62 |
| Ubuntu | 85.20±1.24 | 87.29±0.73 | 07.44±0.76 | 86.97±1.28 |
| Mathoverflow | 86.77±1.69 | 86.83±0.85 | 89.13±0.33 | 89.11±0.90 |
| College | 95.71±0.19 | 95.87±0.22 | 96.04±0.07 | 95.61±0.07 |

**Table 15: Performance of CrossLinkwith varying evolution sequence lengths. Note that here the model is fixed and only varying lengths of trained models. See more details in Table 5.**

| Seq length | 1 | 29 | 59 | 89 | 119 |
|---|---|---|---|---|---|
| Nearby | 82.25 | 85.16 | 86.69 | 88.65 | **89.61** |
| Myket | 81.90 | 87.47 | 87.50 | 87.71 | **87.72** |
| Ubuntu | 71.91 | 79.07 | 78.79 | 84.92 | **87.44** |
| Mathover. | 80.42 | 84.34 | 84.21 | 87.31 | **89.13** |

**Table 16: Performance of CrossLinkwith evolving patterns from other graphs with 119 sequence length. For comparison, we report the performance under the same domain while the sequence length is only 1. See more details in Table 6.**

| | # Seq = 1 | Evolving pattern of other domain | | | |
|---|---|---|---|---|---|
| | | Mooc | LastFM | Review | Reddit |
| Nearby | 82.26 | 80.76 | 82.00 | **87.47** | 84.33 |
| Myket | 81.95 | 79.30 | 86.23 | 86.63 | **87.78** |
| Ubuntu | 71.92 | 75.34 | 74.96 | **88.78** | 79.68 |
| Mathover. | 80.43 | 81.80 | 81.16 | **89.10** | 83.68 |

