# OpenReview forum: "Enhancing Cross-domain Link Prediction via Evolution Process Modeling"
_ACM.org/TheWebConf/2025/Conference — WWW 2025 Poster_

### Official Review · Reviewer_dJJz · 2024-11-26

**Novelty:** 5
**Technical Quality:** 4

**Review:**

This paper studies the problem of cross-domain link prediction over dynamic graphs, which is an interesting and relatively new topic. To address this problem, this paper integrates evolution process modeling into a Transformer to enhance cross-domain link prediction performance. Also, this paper proposes a time augmentation to minimize the disparity for cross-domain learning. Extensive experiments over multiple datasets validate the effectiveness and generalization capability of the proposed method.

**Pos**:

1. Time augmentation is important for cross-domain link prediction over dynamic graphs and the proposed strategy looks good in this task.
2. The improvement of the proposed method is significant.
3. This paper provides several insights based on the experimental observation, making the proposed method clear.

**Cons**:
1. This paper argues that the inference is efficient on page 4, but there is no experimental support for this claim.
2. This paper uses K-V caching in the inference stage, but there are no detailed explanations. What is the function and how about its influence?
3. The main content of this paper can be reorganized to fit within the 8-page limit. In its current form, many results are placed in the appendix, requiring readers to refer to it frequently.

**Questions:**

1. Could authors provide inference time comparison?
2. Although this paper proposes a time augmentation, it is unclear how to process the order of different datasets. How about the influences of the order? And is it one-by-one datasets or one-by-one edge events?
3. Could you please provide the detailed parameter setting of CrossLink?
4. Does the GraphMixer work in the CrossLink framework as the graph encoder? How about the results?

**Reviewer Confidence:**

4: The reviewer is certain that the evaluation is correct and very familiar with the relevant literature

**Scope:**

3: The work is somewhat relevant to the Web and to the track, and is of narrow interest to a sub-community

---

### Official Review · Reviewer_88GF · 2024-11-29

**Novelty:** 5
**Technical Quality:** 4

**Review:**

**Summary**

This paper proposes CrossLink, a new framework designed for predicting links across different domains. CrossLink analyzes the evolution patterns of a specific downstream graph and then makes predictions based on those patterns. It uses a method called conditioned link generation, which combines evolution and structure modeling to focus on evolution-related link predictions. This method is implemented through a transformer-decoder structure, allowing for efficient parallel training and inference. CrossLink is trained using a wide range of dynamic graphs from various domains, 6 million dynamic edges in total. Extensive experiments on eight untrained graphs shows that CrossLink achieves SOTA performance in cross-domain link prediction. When compared to advanced baseline models under the same conditions, CrossLink improves the Average Precision by an average of 11.40% across the eight graphs.

**Strength**

1.CrossLink employs the Graph Encoder to extract the hidden evolution patterns in dynamic graphs and transformer-decoder to enable efficient inference.

2.CrossLink acquires a broader range of knowledge since it is trained on multiple domains,which efficiently provides a novel approach to cross-domain problems.

3. Extensive experiments. The authors have conducted numerous and various experiments to verify the effectiveness of proposed CrossLink.

**Weakness**

1.In 4.2, CrossLink just samples a series of node pairs to represent the whole evolution patterns. It exactly decreases the complexity of the model,but will it also makes the model lose some significant information of the dynamic graph?

2.CrossLink aims to solve cross-domain problems, but the domains/graphs employed in the experiment doesn't hold similarities, which means the domains are isolated.

3.Some spelling mistakes:(1) In Keywords: Link "prediction" instead of "prerdiction"; (2)In 6:"CONCLUSION" instead of "CONCULSION"

**Questions:**

Please see my listed weaknesses.

**Ethics Review Description:**

N/A.

**Reviewer Confidence:**

3: The reviewer is confident but not certain that the evaluation is correct

**Scope:**

3: The work is somewhat relevant to the Web and to the track, and is of narrow interest to a sub-community

---

### Official Review · Reviewer_VTKS · 2024-12-01

**Novelty:** 3
**Technical Quality:** 4

**Review:**

The paper proposes a dynamic graph-based cross-domain link prediction method, which utilizes temporal changes to predict possible future links by introducing temporal information on the graph structure. The authors adopted conditioned link generation (CLG) method, which combines graph adjacency and time series data for training, in an attempt to capture the dynamic interaction information between nodes, thus effectively improving the prediction accuracy.

Strengths:
1. This paper is the first to consider the cross-domain dynamic link prediction scenario, adapting to diverse real-world applications.
2. This paper proposes a transformer-based model, CrossLink, which conducts cross-domain link prediction by integrating evolution process modeling, achieving state-of-the-art performance.

Weakness:
1. This paper does not explain in detail how by modeling the graph evolution process the model acquires the ability to discriminate the ambiguous structures of different graphs. While the paper shows the model's performance on some datasets, it does not specifically test how the evolutionary process of the graph helps in identifying ambiguous structures.
2. This paper emphasizes cross-domain dynamic link prediction scenarios in its writing, but does not focus on the specific challenges associated with the task, making it difficult for readers to grasp the key points.

**Questions:**

Q1. This paper addresses cross-domain dynamic graph link prediction, but whether cross-link maintains its effectiveness when there is significant heterogeneity between the trained and tested graphs (for example, training on a social network and testing on a protein network).

Q2. In the experimental setup section, this paper sets the node features and edge features to 0 (these features tend to significantly affect the heterogeneity between graphs), whether this simplifies the problem and thus affects the effectiveness of cross-link in real-world scenarios.

**Reviewer Confidence:**

3: The reviewer is confident but not certain that the evaluation is correct

**Scope:**

4: The work is relevant to the Web and to the track, and is of broad interest to the community

---

### Official Review · Reviewer_hUbD · 2024-12-01

**Novelty:** 5
**Technical Quality:** 5

**Review:**

This paper introduces a cross-domain link prediction framework named CrossLink. CrossLink enhances the effectiveness of connection prediction by incorporating the evolutionary history of graphs, implemented based on a transformer-decoder architecture, and supports efficient parallel training and inference. CrossLink is trained on dynamic graphs from various domains and demonstrates good generalization, performing superiorly across multiple datasets.

Pros:
- The paper has a clear overall structure and logic, and it introduces the motivation, principles, and implementation of the CrossLink framework in a relatively clear manner.
- It is innovative in introducing evolutionary history to cross-domain link prediction tasks for the first time, representing a progression from static to dynamic graphs.
- The experimental validation section is clear and detailed, accurately showcasing the model's superior performance.

Cons:
- There are layout errors; the symbols on line 248 extend beyond the text block boundaries.
- In the related work chapter, the "Foundation model" section, while somewhat related to the motivation of this paper, has little to do with the specific content of the paper, yet it takes up half of the section.

**Questions:**

The paper presents an approach that enhances link prediction by incorporating the evolutionary history of graphs. While this method may not have been previously employed in cross-domain link prediction tasks, has it been proposed in single-graph link prediction tasks?

In comparison to methods that enhance link prediction by introducing the evolutionary history of graphs in single-graph link prediction tasks, where lies the novelty of the approach proposed in this paper?

**Reviewer Confidence:**

3: The reviewer is confident but not certain that the evaluation is correct

**Scope:**

3: The work is somewhat relevant to the Web and to the track, and is of narrow interest to a sub-community

---

### Official Review · Reviewer_ySED · 2024-12-03

**Novelty:** 5
**Technical Quality:** 3

**Review:**

In this paper, the authors study the problem of cross-domain link prediction on dynamic graphs, and propose to model the evolution of links using decoder-only transformers to enable multi-domain knowledge transfer. Comprehensive experiments are conducted to evaluate the proposed method on several datasets, and the results prove the effectiveness of the proposed method. The analysis of the results also provides insights.

Pros:

1. The problem of interest is important and interesting, and can potentially benefit broad applications.
2. The experiments are extensive and cover a wide range of settings. The experimental results are promising, which shows the effectiveness of the proposed method. The analysis of the results is also insightful.

Cons:

1. My main concern is that the construction of evolving sequences is not explained in detail, which I think is crucial for the effectiveness of the proposed method.
    - How are positive edges sampled?
    - How are negative edges sampled and what is the ratio of negative to positive edges?
    - Are they sampled uniformly or using random walk-like strategies?
    - During inference, are the sampled sequences agnostic to the edges being predicted?
    - The authors mention that the kv-cache can be enabled for efficient inference. Does this mean only one pre-sampled sequence is used for all the edges being predicted?
    - What's the size of training sets in terms of the number of sequences?
2. Reproducibility. The high-level structure of the proposed method is relatively clear, but there are many details. Code is encouraged to be released. Moreover, what's the training setup? Since the proposed model is a 12-layer transformer, with GNNs as token encoders, it might require a lot of computational resources.
3. Figure 1 is a nice illustration of the motivation. However, if this phenomenon holds for real-world datasets? The authors are encouraged to provide some analysis on this, e.g., how heterogeneous the datasets are.
4. What are `w/o transformer` in Fig 3 and `w/o encoder` in Table 8?

Minor comments:
1. Line 452: super-parameters -> hyper-parameters?
2. Table 2: `denoted as 1F5F8 for yes and 2613 for no`, what do these numbers mean?
3. Table 3: Weather -> Whether

**Questions:**

Please refer to Cons.

**Reviewer Confidence:**

3: The reviewer is confident but not certain that the evaluation is correct

**Scope:**

4: The work is relevant to the Web and to the track, and is of broad interest to the community